# Real-Time Monitoring of Critical Quality Attributes during High-Shear Wet Granulation Process by Near-Infrared Spectroscopy Effect of Water Addition and Stirring Speed on Pharmaceutical Properties of the Granules

**DOI:** 10.3390/ph15070822

**Published:** 2022-07-02

**Authors:** Keita Koyanagi, Akinori Ueno, Tetsuo Sasaki, Makoto Otsuka

**Affiliations:** 1Earthtechnica Corporation Limited, 1780 Kamikouya, Yachiyo 276-0022, Japan; koyanagi_keita@earthtechnica.co.jp (K.K.); ueno_aki@earthtechnica.co.jp (A.U.); 2Graduate School of Medical Photonics, Shizuoka University, 3-5-1 Johoku, Naka-ku, Hamamatsu 432-8011, Japan; sasaki.tetsuo@shizuoka.ac.jp; 3Research Institute of Electronics, Shizuoka University, 3-5-1 Johoku, Naka-ku, Hamamatsu 432-8011, Japan

**Keywords:** real-time monitoring, high-shear wet granulation process, near-infrared spectroscopy, median particle size, partial least squares regression analysis, critical quality attribute, pharmaceutical properties of the granules

## Abstract

To produce high-quality pharmaceuticals, a real-time monitoring method for the high-shear wet granulation process (HSWG) was developed based on near-infrared spectroscopy (NIRS). Samples consisting of lactose, potato starch, and hydroxypropyl cellulose were prepared using HSWG with varying amounts of purified water (80, 90, and 100 mL) and impeller speed (200, 400, and 600 rpm), which produces granules of different characteristics. Twelve batches of samples were used for the calibration and nine batches were used for validation. After drying, the median particle size (D50), tapped density (TD), and Hauser ratio (HR) were measured. The best calibration models to predict moisture content (MC), D50, TD, and HR were determined based on pretreated NIR spectra using partial least squares regression analysis (PLSR). The temporal changes in the pharmaceutical properties under different amounts of water added and stirring speed were monitored in real time using NIRS/PLSR. Because the most important critical quality attribute (CQA) in the process was MC, granule characteristics such as D50, TD, and HR were analyzed with respect to MC. They might be used as robust and simple monitoring methods based on MC to evaluate the pharmaceutical properties of HSWG granules.

## 1. Introduction

The pharmaceutical manufacturing process consists of several individual operations, such as mixing, granulation, drying, tableting, and coating, which are controlled under good manufacturing practice (GMP) guidelines by governmental offices. In particular, because the granulation process is considered one of the most important unit operations to produce high-quality medicines, delicate techniques by skilled veteran technicians are required to control the process [1]. The granules produced in the granulation process are used as intermediate products for pharmaceutical products (granules for tableting), and their critical quality attributes (CQA) greatly affect product quality, such as hardness and disintegration time, and dissolution of the tablets as the final products [2]. High-shear wet granulation (HSWG) is a relatively easy granulation method for increasing the particle size to improve powder properties such as bulk density and powder fluidity suitable for raw granules for tableting [3]. HSWG can offer several advantages over other granulation methods, such as excellent mixing effect, short processing time, high drug loading rate, high efficiency, low energy consumption, reduction of process dust, complete closure, etc. [4]. Therefore, the HSWG is applied in a wide range of fields such as pharmaceuticals, pesticides, dietary supplements, foods, chemicals, fertilizers, and electronic materials [5]. Meanwhile, the guidance on process analytical technology (PAT), issued by the U.S Food and Drug Administration (FDA) in 2004 as a modern technology for the constant production of high-quality pharmaceuticals, states that PAT ensures the quality of final products [6]. PAT is defined as “a system for designing, analyzing, and controlling manufacturing by measuring the critical quality and performance attributes of raw materials and processes.” [6]. Regulators are encouraging drug development to adopt “quality by design” (QbD), which is a science-based approach by PAT for ensuring product quality [7]. Therefore, the QbD approach that promotes the manufacture of formulations based on an understanding of the dynamic process is an important and necessary basic concept for the fully automated continuous production of high-quality pharmaceuticals [8,9].

Changes in the pharmaceutical properties of raw materials during granulation are monitored by various types of sensors, and the manufacturing process can be controlled during the progress of the process. Several attempts have been made to control the properties of granules produced with HSWG. Measurements of the torque of a powder blender paddle using a power consumption meter [10,11,12], particle size of wet granules using the focused-beam-reflectance measurement [13,14], acoustic emission of granules using a microphone [15,16,17], and powder flow using a drag flow force sensor [18,19] can be used for various non-destructive and non-contact measurement methods for monitoring the granulation process.

However, since the rapid and non-invasive near-infrared spectroscopy (NIRS) has been officially adopted by pharmacopoeia around the world, it is often implemented as a PAT for measuring the physical and chemical properties of granular pharmaceutical materials prepared using the fluid bed granulation process [20,21,22,23,24]. There have been some reports related to real-time monitoring investigations of HSWG using NIRS, such as on polymorphic transformation of the active pharmaceutical ingredient in granules [25], changes in granule properties [26,27,28,29], and the effect of HSWG operation conditions on tablet properties [30]. However, there are few reports on the molecular-level mechanism of granule formation from a powder mass. Therefore, in a previous study, to clarify the role of intermolecular action of a binding liquid during granulation, the granulation process of fine glass bead powder with water as the simplest model was monitored simultaneously using NIRS and power consumption meter methods and then, the dynamic molecular action of the binding liquid during granulation was analyzed using the multivariable method [31]. Furthermore, in [32], the HSWG process of a standard pharmaceutical formulation powder consisting of starch and lactose was monitored using NIRS and power consumption meter analysis and the dynamic molecular action of the swelling of the starch by absorption of the binding liquid and the dissolution of lactose during granulation were observed in real time. In this study, robust partial least squares regression (PLSR) calibration models to predict the pharmaceutical properties of granules prepared using HSWG were established using the real-time NIRS/chemometrics method and the effects of the water amount, water addition rate, and impeller agitation speed on the pharmaceutical properties of the granules were investigated.

## 2. Results

### 2.1. Establishment of PLSR Calibration Models for Evaluating Pharmaceutical Properties of HSWG Granules by NIRS/PLSR

Granules consisting of the standard formulation powder (LA:PS = 7:3) [33] (total weight: 500 g) with HPC-L were prepared using HSWG under various granulation conditions. The details of the mixture used for calibration models and for prediction by NIRS/PLSR are shown in Table 1 and Table 2, respectively. A set of NIR spectra of granules prepared using HSWG (C#1–12) was used to obtain the calibration models to predict various pharmaceutical properties, such as MC, D50, TD, and HR, as shown in Table 1, and the other NIR-spectra set (P#1–9) was used for factor analysis of the granulation processes. As shown in Figure 1, the granules obtained under the conditions of a higher amount of binding water and higher stirring speed had larger mean particle sizes.

Figure 2 shows the typical changes in the NIR spectra of the C#2 formulation, the details of which are presented in Table 1, during the HSWG process; the peak heights at 5160 and 6780 cm^−1^ increased with increasing amounts of water added as a binder liquid. Subsequently, their base lines shifted up and down with increasing amounts of water added.

Figure 3 shows the NIR spectra of raw powder materials for granulation such as LA, PS, LAH, HPC-L, and pure water. All spectra represent specific absorption peaks due to the chemical structures of the individual powder materials.

The calibration models to predict MC, D50, TD, and HR were established by PLSR analysis based on the NIR spectra obtained during the HSWG processes (C#1–10 and C#12) and their pharmaceutical properties (Table 1). The chemometric parameters of the PLSR calibration models based on various pre-treated NIR spectra are summarized in Table 3, and the best calibration models were selected based on the *PRESS*.

Figure 4 shows the relationship between the predicted and measured values of the best-fitting PLSR calibration models. The best plots of MC and TD were based on the NIR spectra pretreated by SNV; however, those of D50 and HR were based on the spectra treated by NOR. To verify the robustness of the best-fit models, the relationships between the predicted and measured values were evaluated using a cross-validation method and the results are shown in Table 3. According to the cross-validation results, coefficients of determination (*γ*^2^) of the best calibration models were found to be 0.89 or higher. Furthermore, the closed circles in Figure 4 and Table 4 show the validation results of the best models for predicting MC, D50, TD, and HR based on external NIR spectra (C#11). These additional validation results suggested that those PLSR models could predict changes in pharmaceutical properties of HSWG granules using NIRS, since individual calibration models had sufficiently smaller *SEP* values, and the linear plots had *γ*^2^ values of more than 0.8. ** is the best fitted model and * is second best fitted model.

### 2.2. Regression Vector (RV) of the Best Calibration Models to Predict Pharmaceutical Properties of HSWG Granular Products

Figure 5 shows the area-normalized RVs of the best calibration models for predicting the MC, D50, TD, and HR of HSWG granules. Because the individual RVs had characteristic positive and negative peaks based on molecular level information in the NIR spectra of the HSWG granules, their specific peak patterns reflect differences in the physicochemical properties of the HSWG granules. This indicates that the patterns of the PLSR calibration models were linked to the pharmaceutical properties of the HSWG granules. Figure 6 shows the HCA dendrogram of the RV of the PLSR calibration models used to predict the pharmaceutical properties of HSWG granules. The *SI_ab_* values of the RVs of the PLSR calibration models decreased in the order *HR ≈ WC > TD ≥ D50*, indicating that the similarity of their individual molecular mechanisms based on NIR measurements was used to measure the physicochemical properties of the granules.

### 2.3. Real-Time-Monitoring of Pharmaceutical Properties of the Granules during HSWG Process Using NIRS

Figure 7 shows the effect of adding water and increasing stirring speed on real-time MC profiles of the HSWG granules predicted by the NIR/PLSR method. The MC of the granules increased as the amount of water added increased, as shown in Table 2. In the granulation experiments with 80, 90, and 100 mL of water added, the MC profiles of the granules in the range 600–800 s obtained at various stirring rates were superimposed with each other. The maximum MC of the granules with 80, 90, and 100 mL of water added in subsequent granulation processes were approximately 15.5, 17.0 and 18.5%, respectively, and their maximum values increased in proportion to the amount of water added.

Figure 8 shows the effect of adding water and increasing stirring speed on real-time D50 profiles of the granules predicted by the NIRS/PLSR method. The maximum and minimum final D50 values were obtained for the P#7 and P#3 granules, respectively. The D50 of the granules tended to increase with amount of water added in the range of 80–100 mL, except for D50 of the granules obtained at 80 mL and 600 rpm. In the 90 mL water adding amount, the D50-time profiles of the granules at all stirring speeds were superimposed with each other, but those of 80 and 100 mL were not. In 80 and 100 mL, the D50 of the granules were changed depending on the balance between the water content and the rotation speed. The D50 values were affected by the water adding amount and stirring speed, and the result was slightly complicated.

Figure 9 shows the effect of adding water and increasing stirring speed on real-time TD profiles of the granules predicted by the NIRS/PLSR method. The TD of the granules increased as the amount of water added increased. The TD profiles for most granular conditions with 80, 90, and 100 mL of water added were superimposed. However, the TD of the granules of P#9 at 100 mL and 200 rpm conditions was excluded, it may be related to the result of their D50 of P#9. Initially, the TD profile of the granules obtained at various stirring speeds could be superimposed on each other; however, in the later granulation stages, the TD increased in proportion to the difference in the amount of added water.

Figure 10 shows the effect of adding water and increasing stirring speed on real-time HR profiles of the granules predicted by the NIRS/PLSR method. For the granulations with 80, 90, and 100 mL of water added, almost all HR profiles of the granules obtained at various stirring rates were superimposed. The minimum HR of the granules for 80, 90, and 100 mL of water added were approximately 1.16, 1.12, and 1.09, respectively. Initially, the HR profile of the granules obtained at various stirring speeds could be superimposed on each other; however, in the later granulation process, the HR decreased in proportion to the difference in the amount of water added.

### 2.4. Effect of CQA of Granulation Process on Critical Product Quality of the HSWG Granules

Figure 11 shows the effect of adding water and increasing stirring speed on the relationship between D50 and MC of the HSWG granules. In general, the D50 of the HSWG granules increased with increasing MC; however, the D50/MC profiles at different stirring rates changed depending on the amount of water added to the granules in the cases of P#7 and P#9.

Figure 12 shows the effect of adding water and increasing stirring speed on the relationship between the TD and MC of the HSWG granules. In all HSWG experiments, the TD of the granules decreased with increasing water addition, and the D50/MC profiles of the granules in the range 0–14% MC obtained at various stirring speed were superimposed with each other. However, in the range 14–18%, the D50/MC profiles at different stirring speeds changed depending on the amount of water added. Specially, the TR of the P#7 case (100 mL, 200 rpm) increased significantly with MC at 15% MC or more.

Figure 13 shows the effect of adding water and increasing stirring speed on the relationship between the HR and MC of the HSWG granules. Under all granulation conditions, the HR gradually decreased as the MC increased, and almost all profiles were superimposed well.

### 2.5. Quantitative Relationship between Final Pharmaceutical Properties of HSWG Granules and Critical Process Parameters

Figure 14 shows three-dimensional plots of pharmaceutical properties of the final HSWG products predicted by the NIR/PLSR method versus their critical process parameters. The D50 (Figure 14a) and TD (Figure 14b) changed significantly depending on the amount of water added and the stirring speed, and larger the amount of water added and the faster the stirring speed were, the larger the D50 and TD were. The HR (Figure 14c) changed significantly depending on the amount of water added; however, the effect of stirring speed was not significant. The HR increased as the amount of water added increased.

## 3. Discussion

### 3.1. Evaluation of Pharmaceutical Properties of HSWG Granules by NIR/PLSR Method

During the manufacturing of granules for tableting, granule characteristics such as average granule size, particle size distribution, powder fluidity, apparent density, granule shape, granule hardness, and residual water content of the manufactured granules are significantly changed by the delicate control of operating conditions. Furthermore, it is well-known that these granule properties have a great influence on pharmaceutical properties, such as tablet hardness, disintegration time, and drug release rate, of the final product tablets [34]. For this reason, real-time monitoring of the granule manufacturing process is important for the mass production of high-quality pharmaceutical products, and NIRS and Raman spectroscopy [35] are practically applied in process monitoring. NIRS can simultaneously extract chemical information, such as residual water content, of pharmaceutical compositions, such as bulk drugs and pharmaceutical additives, as well as physical information, such as particle size and particle size distribution, from the product dosage forms in real time [36]. In particular, the water behavior in oral solid dosage forms during the granulation process analyzed using NIRS is recognized as the target CQA for high-quality pharmaceutical manufacturing. Therefore, in this study, to reproduce granular production with large variations in pharmaceutical properties, the HSWG processes were performed by controlling the amount of water added, their addition patterns, and the stirring speed, as shown in Table 1. As shown in the NIR spectral changes (Figure 2) of the C#2 formulation (100 mL, 600 rpm) during the HSWG process as a typical example, the absorption peaks at 5200 cm^−1^ and 6800 cm^−1^ increased with an increase in the amount of water added, and at the same time their spectral baseline shifted up and down significantly. Spectral baseline shifting is related to the light scattering of the wet powder mass due to granulation [32,37].

To evaluate the pharmaceutical properties of the granules, robust calibration models were established based on the NIR spectra measured under various granulation conditions and MC, D50, TD, and HR as the objective variables, using the PLSR method [25,26,27,28,29,30,31]. After pretreatment with various functions, the chemometrics parameters of the PLSR calibration models were calculated to predict the pharmaceutical properties of the granules; they are summarized in Table 3. MC and TD (open triangle in Figure 4) were able to obtain the most linear calibration models with the lowest standard error of cross validation (SECV) when the NIR spectra were pretreated with SNV. However, the best linear calibration models for D50 and HR were based on NOR pretreated spectra. In addition, the results of verifying each optimal PLSR calibration model using C#11 as external validation data (closed circle in Figure 4 and Table 4) indicated that the best-fitted PLSR calibration models had sufficiently low errors, high linearity, and robustness. The accuracy for the PLSR calibration models decreased in the order MC > HR ≥ D50 ≈ TD. It could be reasoned that the model for MC had the best accuracy because NIRS directly measured the binding water amount due to absorption at 6800 and 5150 cm^−1^ (Figure 2). On the other hand, other parameters were indirectly predicted based on physicochemical phenomena, such as reducing granule size (Figure 1), which was caused by the interaction of binding water between particles in the powder bed [31,32]. However, all calibration models were sufficiently accurate to predict the pharmaceutical properties of the HSWG granules in real time.

Moreover, in order to clarify the scientific evidence on the ability of PLSR calibration models to predict the pharmaceutical properties of the granules, the RV of the models was developed by area-normalized pretreatment and compared with each other (Figure 5). The area-normalized RV of the best calibration model to predict MC had significantly negative sharp peaks at 71,355,299, and 4983 cm^−1^. The peaks of the RV were tightly linked with the absorption peaks due to binding water. As shown in the HSWG process of a similar mixture of LAH and PS in a previous report [32], the RV of the PLSR calibration model to predict MC represented a significant increase in peak intensity at approximately 7150 and 5200 cm^−1^ due to an increase in the amount of water added. The RV of the model for predicting MC was more similar to that for predicting HR, as the latter involved the peaks at 71,355,299, and 4983 cm^−1^; however, RVs corresponding to D50 and TD did not involve these peaks and, hence, were not similar to that corresponding to MC. The results suggest that the calibration models of MC and HR might be more directly dependent on the amount of water added, but those of D50 and TD might not be. However, it is not easy to clearly evaluate the similarity using visible observations of the RVs of these calibration models. Therefore, to evaluate the similarity of the multivariable data, HCA, known as the simplest numerical classification method [38,39] in various academic fields, was applied to evaluate the similarity of RVs. The multivariate distance between pairs of sample groups was calculated and compared, and the complicated data were classified into clusters based on similar attributes. In our previous study [40], the similarity index was evaluated using HCA to quantitatively evaluate the similarity of the PLSR calibration model that assesses the physicochemical properties of ground atorvastatin calcium hydrate. The HCA dendrogram of the RV of the calibration model assessing the properties of the HSWG granules (Figure 6) showed that the RVs of HR and MC were the most similar, and the RVs of HR and D50 were the least similar. Since MC and TD were separated over a relatively long distance, the RV similarity decreased in the order *HR ≈ WC > TD ≥ D50.* The results of the quantitative evaluation of RV similarity by HCA analysis were almost the same as the results of the qualitative judgment by visual inspection of the RV of each calibration model.

### 3.2. Actual Real-Time Monitoring for Pharmaceutical Properties of the Granules during HSWG Process by NIR/PLSR

In the previous section, we described how robust PLSR calibration models were constructed based on the NIRS method, which could be used to evaluate the pharmaceutical properties of granule formulations, such as MC, D50, TD, and HR, in real time during the HSWG process (Figure 4, Table 3 and Table 4).

An actual HSWG was performed on the standard formulation sample under various granulation conditions (Table 2); the in-line monitoring of the processes was carried out using the NIRS method and their pharmaceutical properties were evaluated using the robust PLSR calibration models.

For the evaluation of the MC of the granules (Figure 7), the MC-time profiles of the process under different amounts of added water and a constant stirring speed were continuously measured in real time; thus, they were independent of the stirring conditions. NIRS directly measured the MC of the sample granules due to peaks at approximately 7150 and 5200 cm^−1^, owing to the stretching vibration of the OH group, as shown in the RV of the calibration model (Figure 5).

In contrast, the D50 depended on both the amount of water added and the stirring speed (Figure 8), and the D50-time profiles fluctuated depending on the granulation conditions and were unstable. However, in the case of 90 mL of water added, the profiles were superimposed regardless of the stirring conditions. This result indicated that a stable product with a uniform size distribution was maintained during the granulation process when the amount of water added was limited to approximately 90 mL.

The TD-time profiles after adding water could almost be continuously monitored independent of the stirring conditions (Figure 9) of the granulation processes at 80 and 90 mL. However, the profiles at 100 mL changed depending on the stirring speed, and the granulation processes were unstable. In particular, at 200 rpm, TD exhibited a large fluctuation.

Additionally, the HR-time profiles were almost dependent on the amount of added water and were independent of the stirring conditions (Figure 10). This result suggests that the HR of the granules remained stable during granulation under all stirring speeds when the amount of water added was controlled. This means that the characteristics of HR were independent of the stirring condition and more similar to those of MC and not to those of D50 and TD.

All pharmaceutical properties of the HSWG granules, such as D50, TD, and HR, were secondary characteristics of the granules caused by the binding action of water during granulation. However, the degree of dependency of the properties on the amount of water added was not equal in all cases; the dependency decreased in the order of: HR > TD > D50. Moreover, it is interesting that this order might almost be the same as that of the HCA *SI_ab_* of the calibration-model RV (Figure 6).

### 3.3. CQA Monitoring for Controlling Critical Product Quality of HSWG Granules by NIR/PLSR

The current good manufacturing practice (cGMP) guidelines proposed by the FDA for pharmaceutical manufacturing [6,7] recommend monitoring evidence-based pharmaceutical process control after grasping the most important CQA. Therefore, many papers have been reported [10,11,12,13] on PAT for granular manufacturing of solid pharmaceutical dosage forms. In particular, to control wet granulation [20,21,22,23,24,25,26,27,28,29,30,31,32] during the manufacture of granules and tablets, the water distribution during the process was recognized as one of the most important key parameters, CQA; therefore, the process is controlled in conventional GMP manufacturing by skilled engineers.

In this paper, therefore, the MC of the granules as CQA was evaluated in real time during the HSWG process using the NIRS/PLSR method [41], and at same time, the other related pharmaceutical properties were also evaluated. The tendencies shown by all pharmaceutical property-time profiles during the HSWG process are indicated in Figure 7, Figure 8, Figure 9 and Figure 10. Since the pharmaceutical properties of the granules, such as D50, TD, and HR, might change depending on the MC of the dosage forms (the most important CQA), the dependency of these properties on MC were evaluated (Figure 11, Figure 12 and Figure 13).

In general, there are two attractive forces acting between the powder particles that increase the particle size in the HSWG process. One is due to the surface tension of water as the binding liquid [33] and the other is the centrifugal force (CF) in Equation (1), which increases in proportion to the increase in the rotation speed of the impeller [42]. In contrast, a dispersion force that increases with the chopper rotation speed reduces the particle size.
(1)$$CF=mr\omega^{2}=m\frac{v^{2}}{r}\;$$
where *m* denotes internal mass; *r* denotes radius; *v* denotes velocity; and *ω* denotes the angular velocity.

The particle size of the granules depends on the surface tension of the water; it is the driving force that aggregates the powder. Particle size is also controlled by the cohesive force due to the centrifugal force and the dispersion force due to shearing by a chopper. Therefore, the particle size is determined by the total particle cohesiveness owing to the balanced forces. In the present study, HSWG was performed under simple experimental conditions; the impeller rotation speed was varied, and the chopper rotation speed was maintained at 2000 rpm. However, the dispersion force exerted by the chopper was partially independent of the impeller rotation speed.

To properly understand the granulation mechanism of HSWG, real-time changes in the pharmaceutical characteristic values of the granules (Figure 7, Figure 8, Figure 9 and Figure 10) were analyzed; the results indicated that these phenomena might be dependent on the distribution of water. Therefore, to clarify the mechanisms related to water, the monitoring data profiles were analyzed with respect to the MC (CQA) (Figure 11, Figure 12 and Figure 13). The pharmaceutical property–MC profiles indicated the dependency of the properties on the amount of water added. When monitoring the HSWG process, the introduction of pharmaceutical properties–MC profiles might be useful for understanding the process mechanism. There is a possibility that the process could be simplified; changes in the process could be quantitatively monitored, and design space for manufacturing a product that conforms to the formulation standard could be set more easily.

## 4. Materials and Methods

### 4.1. Materials

*α*-Lactose anhydride (LA, used as a diluent) was obtained from DFE Pharma (Corporate Head Office, Goch, Germany). *α*-Lactose monohydrate (LAH, used as a diluent) was obtained from FUJIFILM Wako Chemicals Co. (Tokyo, Japan). Potato starch (PS, as a disintegration agent) was obtained from Kosakai Pharmaceutical Co., Ltd. (Nagaoka, Japan). Hydroxypropyl cellulose (HPC-L; lot no. NEA-4131, used as a binder agent) was obtained from Nippon Soda Co., Ltd. (Tokyo, Japan). The raw powder sample consisting of the standard formulation powder (LA:PS = 7:3) [33] was added with 4% *w*/*w* of HPC-L, and total powder weight was 500 g.

### 4.2. Granulation

The HSWG instrument (Type FS 2, high speed mixer, 2.0 l in internal volume, Earthtechnica Co., Ltd., Tokyo, Japan) with NIR diffuse fiber-optic probe was used for granulation. A total of 500 g of the sample powder was placed in a chamber and mixed using the impeller at 200, 400, and 600 rpm and the chopper at 2000 rpm for 10 min. Next, without stopping the apparatus, purified water (80, 90, and 100 mL) was added to 5, 10, or 20 mL of purified water from the injector, following the conditions in Table 1 (C#1–12), and the mixture was stirred for a predetermined time. All the granulation operations were repeated twelve times under various conditions. Approximately 10 g of granular samples were collected from the sampling port of the granulator at predetermined intervals during the calibration process. The collected wet granules were dried in a tray dryer (SPH-201, Ozawa Science Co., Ltd., Nagoya, Japan) for more than 18 h at 60 °C; its weight loss was measured using an electronic balance (ATX224R, Shimadzu Co. Ltd., Kyoto, Japan), and the moisture content of the granular samples at time *t* during the granulation process was defined as MC. Furthermore, the granulation process for mechanism analysis of the HSWG was repeated nine times according to the operating conditions listed in Table 2 (P#1–9), and the NIR spectra were obtained.

### 4.3. Evaluation of Micrometrics of Granules

After drying, the particle size of the collected sample granules was measured using a sieving test (refer to Japanese Pharmacopeia 18) [43]. Nine sieve screens (Mesh No. 18, 22, 30, 42, 70, 100, 140, and 200 with 850, 710, 500, 355, 212, 150, 106, and 75 μm apertures, respectively; Testing Sieve, Tokyo Screen, Tokyo, Japan) were set up for particle size analysis. After 5 g of granule sample was transferred to pre-weighed sieves and shaken on a vibrator for 15 s, the mass fractions of the granules were measured by weighing. Particle size corresponding to 50% on the cumulative weight percent curve of the granular sample was evaluated as median particle size (D50). The loose powder density (LD) and tapped density (TD) of the dried sample granules were measured as follows. Five grams of sample granules were gently placed into a graduated cylinder (2 cm in diameter and 50 mL in volume) and its volume was measured, and then, LD was evaluated based on volume and weight. Next, the cylinder was tapped one hundred times by hand. TD was evaluated from the sample volume after its measurement following the tapping test (Japanese Pharmacopeia 18). Hauser ratio (HR) was evaluated based on Equation (2) as an index of the powder flowability of the sample [44]. The average of three measurements was taken as the experimental values for the granular samples.
(2)$$HR=\frac{TD}{LD}$$

### 4.4. Measurement of NIRS

The NIR spectra were obtained in the range of 10,000–4000 cm^−1^ with a resolution of 16 cm^−1^ in 48 scans using an NIRS instrument (MPA, Bruker Optics, Ettlingen, Germany) with a diffused fiber-optic probe. The spectral datasets for the establishment of the calibration model and its validation were measured every 20 s for 30 min during the HSWG process under various experimental conditions, as shown in Table 1. To obtain the calibration models to predict the MC, D50, TD, and HR of the dried granules during the HSWG process, 185 NIR spectra obtained were divided into 150 spectra for the calibration data set and 35 spectra for external-validation data set, labeled as C#1–12. The NIR spectra were pretreated with various functions, including the first derivative (1st), second derivative (2nd), area normalization (NOR), multiplicative scatter correction (MSC), and standard normal variate (SNV) [38]. The best calibration models for predicting the various properties (MC, D50, TD, and HR) of the samples were determined based on 115 pretreated NIR spectra (C#1–10 and C#12) using the leave-one-out cross-validation method (LOOCV) with PLSR analysis in the chemometrics software, Pirouette (Ver. 4.5, Infometrix Co., Woodenville, WA, USA) [39].

The original descriptions of multivariate PLSR [38,39] is
(3)$$X=TP^{T}+E$$
(4)$$Y=UQ^{T}+F$$
where *X*, *n × m* matrix of predictors; *Y*, *n × p* matrix of responses; *T* and *U*, *n × m* matrix of the *X* score and the *Y* scores, respectively; *p* and *Q*, *m × l* and *p × l* orthogonal loading matrices, respectively; E and F, matrices of the error terms. The decompositions of *X* and *Y* are made so as to maximize the covariance between *T* and *U*. Multivariable analytical methods construct estimates of the linear regression between *X* and *Y* are as follows:(5)$$Y=X\tilde{B}+{\tilde{B}}_{0}$$

$\tilde{B}\;\mathrm{and}\;{\tilde{B}}_{0}$, *n × m* matrix of related regression vector (RV) and errors, respectively.

The optimum number of latent variables (LV) was determined based on the minimum value of the predicted residual error sum of squares (*PRESS)* in the plot of PRESS versus LV, as shown in Equation (6).
(6)$$PRESS=\sum_{i=1}^{n}{\left({\hat{y}}_{i}-y_{i}\right)}^{2}$$
where *ŷ_i_* and *y_i_* are the predicted and reference values, respectively. The quality of the calibration model was assessed in terms of the standard error of prediction (*SEP*), also called the root mean square error of prediction.
(7)$$SEP=\sqrt{\frac{\sum_{i=1}^{n}{\left({\hat{y}}_{i}-y_{i}\right)}^{2}}{n}}$$

SEC is the standard error of calibration.

All the fitted calibration models were validated based on external NIR spectra datasets (C#11) obtained from independent experiments (Table 1). The pharmaceutical properties of the granules during HSWG were evaluated based on 486 NIR spectra (P#1–9) obtained following the experimental conditions presented in Table 2.

Hierarchical cluster analysis (HCA) [38,39] was performed based on the standard scale of similarity, *SI_ab_*, calculated from the multivariate distance between two samples using chemometrics software (HCA, Pirouette 4.5, Infometrix Inc., Bothell, WA, US).
(8)$$d_{ab}=j{\left[\sum_{j}^{m}{\left(X_{aj}-X_{bj}\right)}^{2}\right]}^{1/2}$$
(9)$$SI_{ab}=1-\frac{d_{ab}}{d_{max}}$$
where *d_max_* is the largest distance in the data and *X* is the data matrix of the samples. All data are presented as average of three different measurement values with standard deviations (SD). Data were analyzed using the one-way analysis of variance (ANOVA) at a significance level of 0.05.

## 5. Conclusions

For continuous production in the automatic-manufacturing of high-quality pharmaceutical products in the future, it is necessary to continuously monitor the important CQA of pharmaceutical products using non-destructive and non-contact analysis methods during the manufacturing process. In this study, robust PLSR calibration models were constructed based on NIRS during HSWG process and they could accurately predict the MC, D50, TD, and HR of granules. Granules with various characteristics were produced by varying the amount of water added and stirring speed, and the temporal changes in the pharmaceutical characteristics of the granules were monitored in real time using the NIRS/PLSR method. Because the most important characteristic (CQA) in the granulation process was the distribution of the MC, it was possible to continuously and quantitatively measure the MC and evaluate these in real time to understand granule characteristics such as D50, TD, and HR. The pharmaceutical properties might be understood as a function of MC, in a specific design space, respectively. It has been shown that these models could be constructed as a robust and simple monitoring and evaluation method based on MC as CQA to evaluate the pharmaceutical properties of the HSGW granules.

## Figures and Tables

**Figure 1 pharmaceuticals-15-00822-f001:**
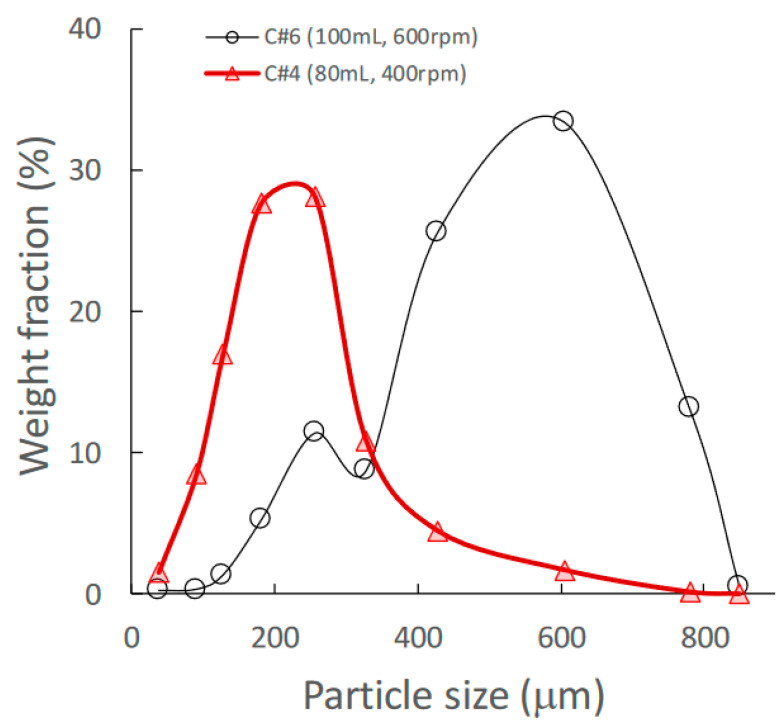
Effect of change in amount of water added on the granular size distribution of high-shear wet granules.

**Figure 2 pharmaceuticals-15-00822-f002:**
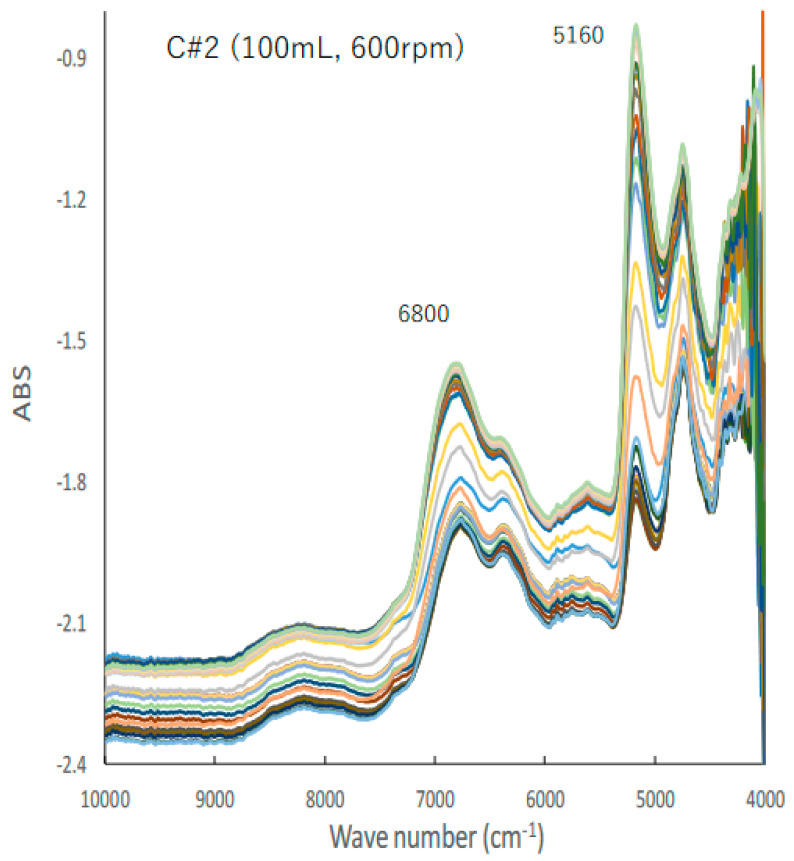
A typical change in the NIR spectra of the C#2 formulation during HSWG process.

**Figure 3 pharmaceuticals-15-00822-f003:**
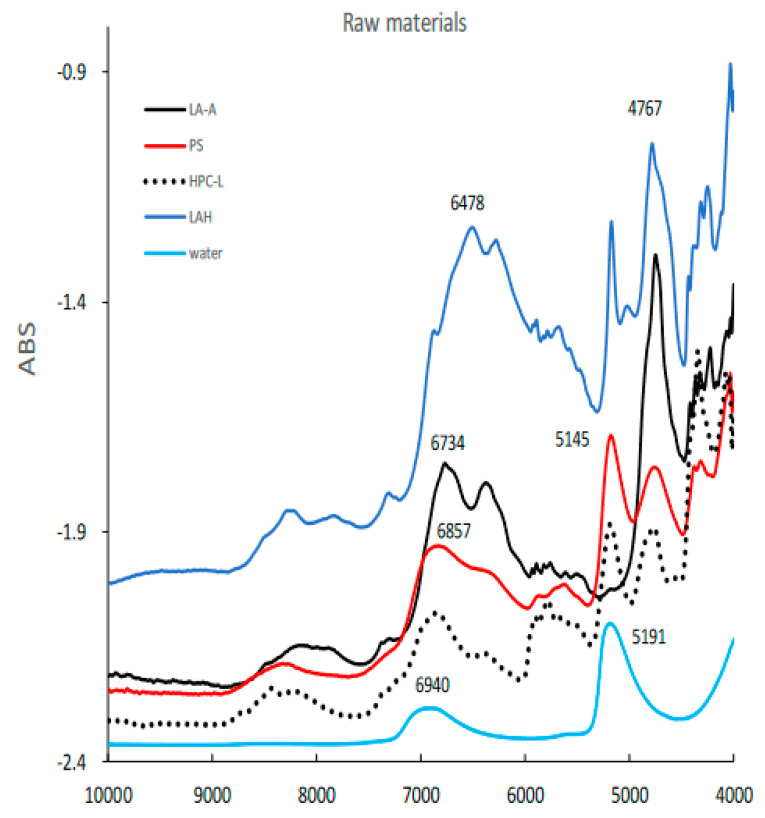
The NIR spectra of raw powder materials for HSWG. LAA, *α* -lactose anhydride; LAH, *α*-lactose monohydrate; PS, potato starch; HPC-L; hydroxypropyl cellulose.

**Figure 4 pharmaceuticals-15-00822-f004:**
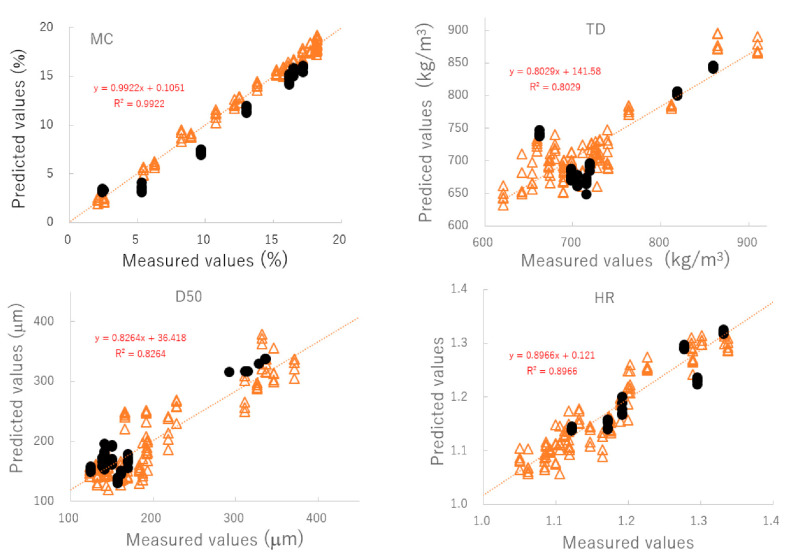
The relationships between the predicted and measured values of the best-fitting PLSR calibration models. MC: moisture content, D50: median particle size, TD: tapped density, HR: Hauser ratio.

**Figure 5 pharmaceuticals-15-00822-f005:**
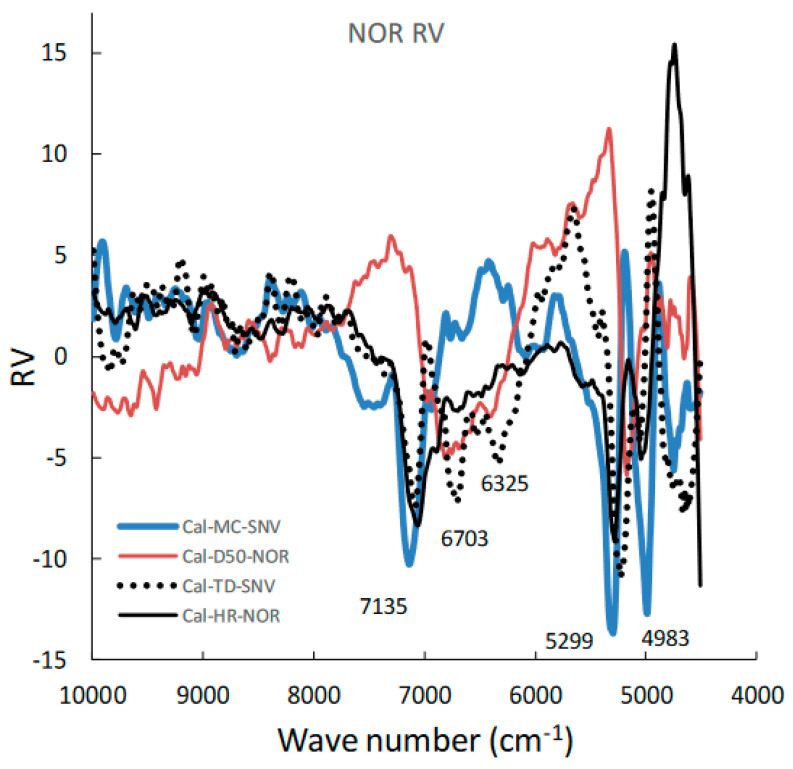
Area-normalized RVs of the best calibration models for predicting the MC, D50, TD, and HR during the HSWG process using NIRS/PLSR.

**Figure 6 pharmaceuticals-15-00822-f006:**
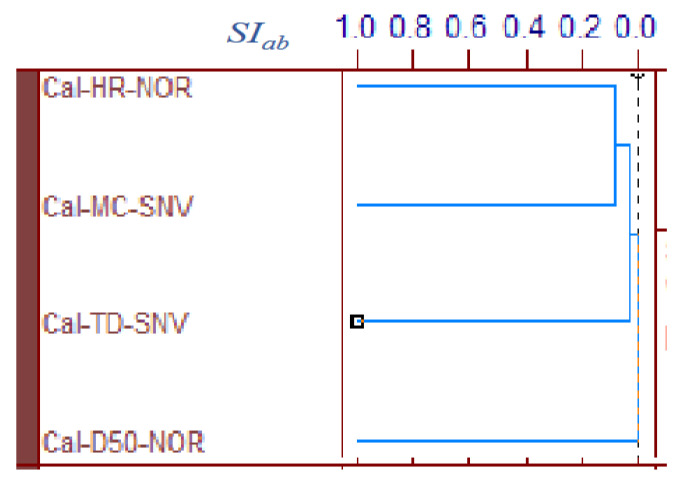
HCA dendrogram of the RVs of the best calibration models to predict the pharmaceutical properties of the HSWG granules. HCA: Hierarchical cluster analysis, *SI_ab_*: the standard scale of similarity.

**Figure 7 pharmaceuticals-15-00822-f007:**
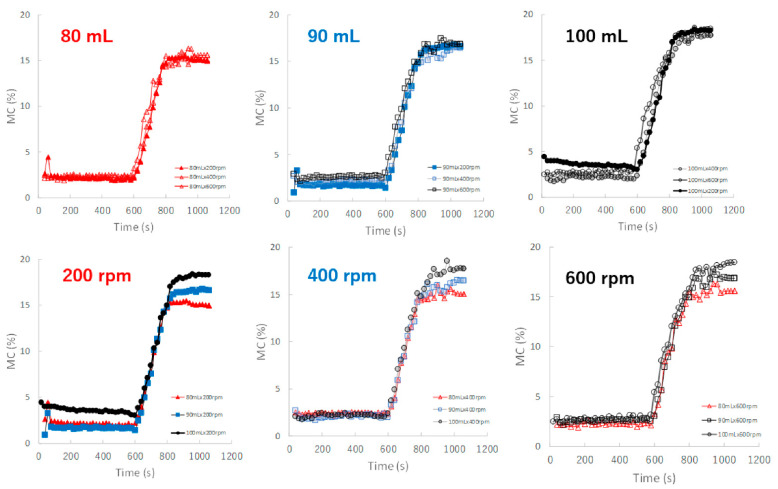
The effect of adding water and increasing stirring speed on real-time MC profiles of the HSWG granules predicted using the NIR/PLSR method.

**Figure 8 pharmaceuticals-15-00822-f008:**
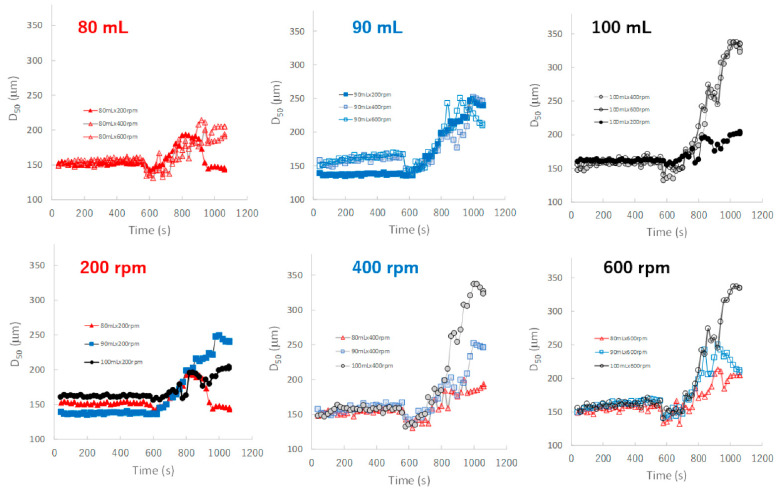
The effect of adding water and increasing stirring speed on real-time D50 profiles of the granules predicted using the NIRS/PLSR method.

**Figure 9 pharmaceuticals-15-00822-f009:**
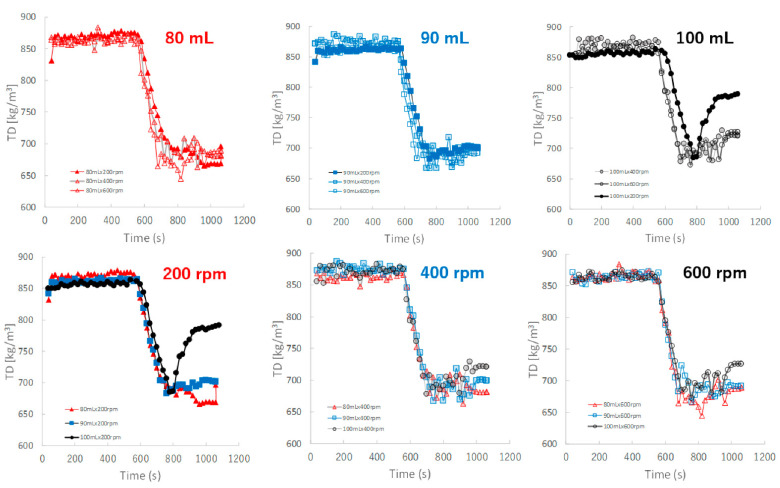
The effect of adding water and increasing stirring speed on real-time TD profiles of the granules predicted using the NIRS/PLSR method.

**Figure 10 pharmaceuticals-15-00822-f010:**
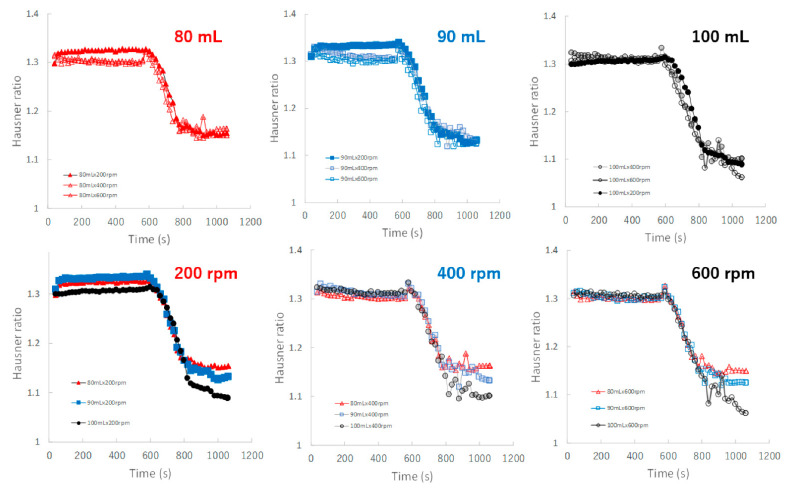
The effect of adding water and increasing stirring speed on real-time HR profiles of the granules predicted using the NIRS/PLSR method.

**Figure 11 pharmaceuticals-15-00822-f011:**
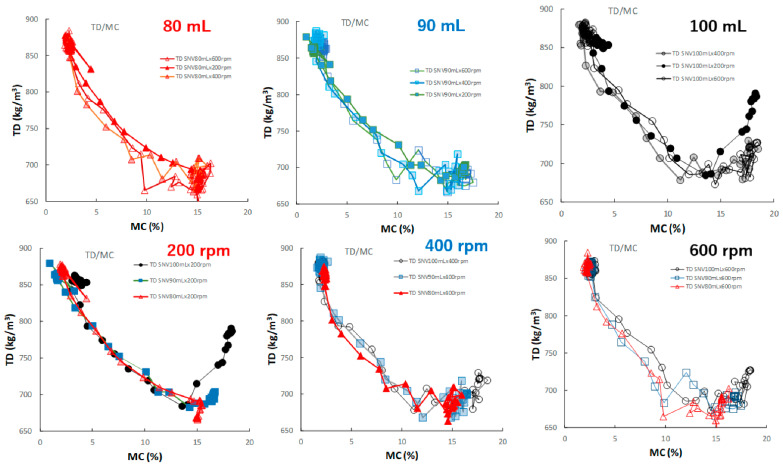
The effect of adding water and increasing stirring speed on the relationship between D50 and MC of the HSWG granules.

**Figure 12 pharmaceuticals-15-00822-f012:**
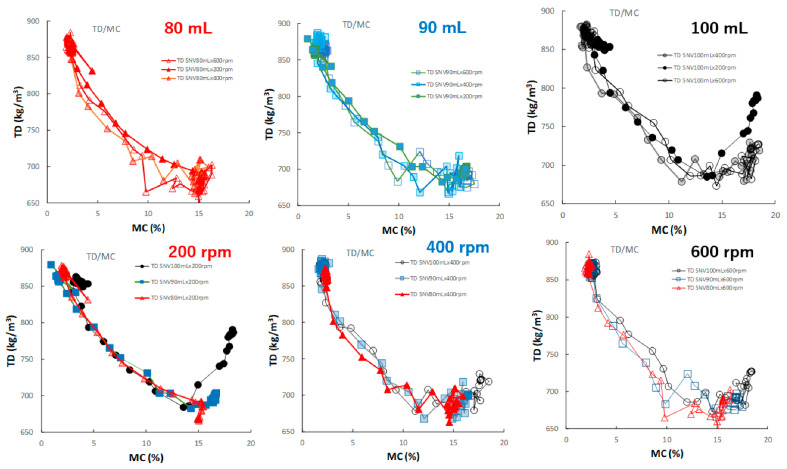
The effect of adding water and increasing stirring speed on the relationship between TD and MC of the HSWG granules.

**Figure 13 pharmaceuticals-15-00822-f013:**
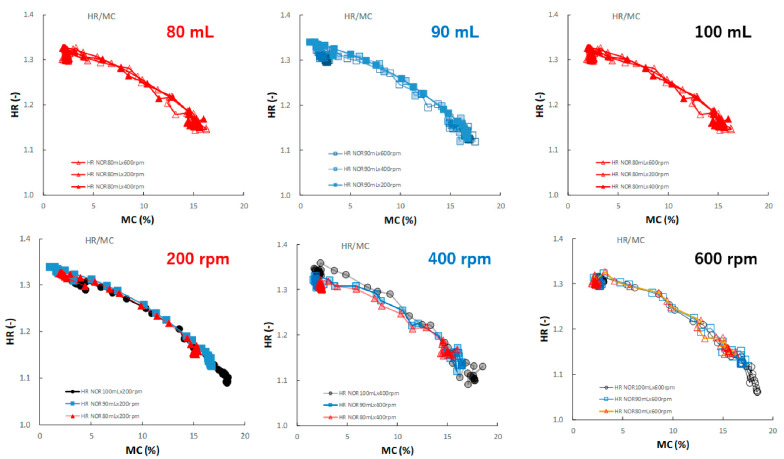
The effect of adding water and increasing stirring speed on the relationship between HR and MC of the HSWG granules.

**Figure 14 pharmaceuticals-15-00822-f014:**
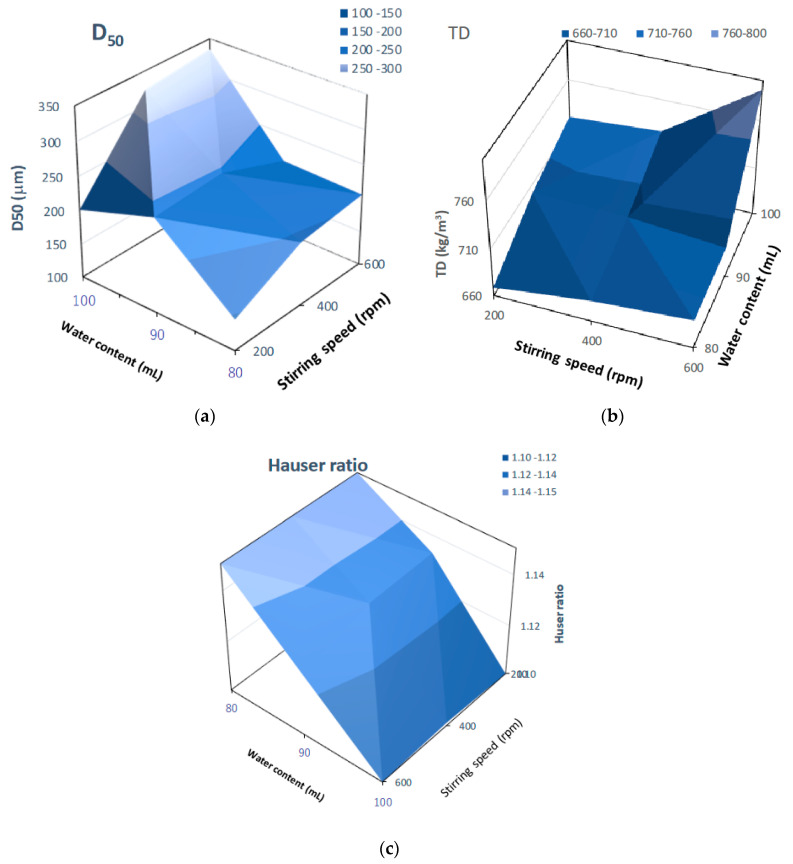
Three-dimensional plots of pharmaceutical properties of the final HSWG products predicted using the NIR/PLSR method versus their critical process parameters. (**a**) D50; (**b**) TD; (**c**) HR.

**Table 1 pharmaceuticals-15-00822-t001:** Preparation and various pharmaceutical properties of HSWG granules for the calibration model.

No	1	2	3	4	5	6	7	8	9	10	11	12
Powder mixture (g)	500	500	500	500	500	500	500	500	500	500	500	500
Time (s)	Adding water amount (mL)											
600	20	20	20	20	20	20	20	20	20	20	20	20
660	20	20	20	20	0	0	0	20	20	20	20	20
720	20	20	20	20	20	20	20	20	20	20	20	20
780	20	20	20	20	0	0	0	20	20	20	20	20
840	20	20	10	0	20	20	10	10	20	20	10	20
960	0	0	10	0	20	20	10	0	0	0	0	0
1020	0	0	0	0	5	5	10	0	0	0	0	0
1200	0	0	0	0	0	5	10	0	0	0	0	0
1320	0	0	0	0	0	5	0	0	0	0	0	0
1440	0	0	0	0	0	5	0	0	0	0	0	0
Total	100	100	100	80	85	100	80	90	100	100	90	100
Stirring (rpm)	600	600	200	400	600	600	400	400	400	400	200	200
Chopper (rpm)	2000	2000	2000	2000	2000	2000	2000	2000	2000	2000	2000	2000
MC [%]	18.1	18.2	16.5	15.8	16.1	18.2	15.8	17.1	18.1	18.2	17.2	18.2
D50 [μm]	337	346	166	142	218	332	161	192	371	311	170	193
TD [kg/m^3^]	721	739	720	702	654	659	621	690	726	731	716	812
HR [-]	1.08	1.05	1.11	1.17	1.11	1.06	1.12	1.18	1.09	1.09	1.12	1.12

**Table 2 pharmaceuticals-15-00822-t002:** Preparation of HSWG granules for analyzing the granulation mechanism under various experimental conditions.

No	1	2	3	4	5	6	7	8	9
Powder mixture (g)	500	500	500	500	500	500	500	500	500
Time (s)	Adding water amount (mL)								
600	20	20	20	20	20	20	20	20	20
660	20	20	20	20	20	20	20	20	20
720	20	20	20	20	20	20	20	20	20
780	20	20	20	20	20	20	20	20	20
840	20	10	0	20	10	0	20	10	0
Total	100	90	80	100	90	80	100	90	80
Stirring (rpm)	600	600	600	400	400	400	200	200	200
Chopper (rpm)	2000	2000	2000	2000	2000	2000	2000	2000	2000

**Table 3 pharmaceuticals-15-00822-t003:** Chemometrics parameters of the PLSR calibration models to predict pharmaceutical properties of the HSWG granules based on various pre-treated NIR spectra.

**MC**	**PTM**	** *LV* **	** *CPV* **	** *SECV* **	** *PRESS Val* **	** *γ* ^2^ *Val* **	** *SEC* **	** *PRESS Cal* **	** *γ* ^2^ *Cal* **	
	1st	5	91.0	0.953	109	0.983	0.836	79.6	0.987	
	2nd	6	75.5	1.164	162	0.974	0.815	75.0	0.988	
	MCS	5	83.5	0.687	56.6	0.991	0.490	27.4	0.996	*
	NOR	5	96.4	0.838	84.2	0.987	0.620	43.8	0.993	
	SNV	5	83.1	0.673	54.4	0.991	0.465	24.6	0.996	**
**D50**	**PTM**	** *LV* **	** *CPV* **	** *SECV* **	** *PRESS Val* **	** *γ* ^2^ *Val* **	** *SEC* **	** *PRESS Cal* **	** *γ* ^2^ *Cal* **	
	1st	4	90.9	49.4	293400	0.777	47.2	256121	0.808	
	2nd	4	66.0	59.3	421436	0.669	51.4	303263	0.767	
	MCS	4	82.1	44.8	240431	0.824	38.2	167447	0.879	
	NOR	4	96.1	39.5	187381	0.864	33.4	128083	0.909	**
	SNV	4	81.6	44.7	240091	0.824	38.1	167006	0.880	*
**TD**	**PTM**	** *LV* **	** *CPV* **	** *SECV* **	** *PRESS Val* **	** *γ* ^2^ *Val* **	** *SEC* **	** *PRESS Cal* **	***γ*^2^ *Cal***	
	1st	1	66.0	60.5	439611	0.370	59.2	414126	0.429	
	2nd	1	30.5	60.6	440422	0.370	58.7	407014	0.445	
	MCS	1	76.4	56.5	383096	0.497	55.2	359240	0.541	
	NOR	6	96.6	48.9	287250	0.683	32.0	115478	0.879	*
	SNV	4	80.2	45.0	242956	0.737	29.5	100087	0.896	**
**HR**	**PTM**	** *LV* **	** *CPV* **	** *SECV* **	** *PRESS Val* **	** *γ* ^2^ *Val* **	** *SEC* **	** *PRESS Cal* **	** *γ* ^2^ *Cal* **	
	1st	4	90.7	0.0331	0.132	0.920	0.0301	0.104	0.937	
	2nd	4	66.8	0.0398	0.190	0.884	0.0322	0.119	0.928	
	MCS	2	79.5	0.0299	0.108	0.935	0.0281	0.0926	0.945	
	NOR	3	96.0	0.0303	0.110	0.934	0.0277	0.0890	0.947	**
	SNV	2	78.9	0.0299	0.107	0.936	0.0281	0.0922	0.945	*

*LV* is the number of latent variables; *CPV* is the cumulative percent variable; *SECV* is the standard error of cross-validation; *PRESS Val* is the predicted residual error sum of squares based on validation; *PRESS Cal* is the prediction residual error sum of squares based on calibration; *SEC* is the standard error of calibration; *γ*^2^
*Val* is the coefficient of determination based on cross-validation; and *γ*^2^
*Cal* is the coefficient of determination for calibration. Pretreatment (PTM) of the NIR spectra was conducted using various functions: 1st is the first derivative, 2nd is the second derivative, multiplicative scatter correction, area normalization, and standard normal variate. ** is the best fitted model and * is second best fitted model.

**Table 4 pharmaceuticals-15-00822-t004:** Validation result of the best-fitting PLSR calibration models to predict pharmaceutical properties of the HSWG granules based on external NIRS data sets. *SEP* is the standard error of prediction; *γ*^2^ is the coefficient of determination based on the external validation data; slope and intercept are the slope and y-intercept of the regression line in the plots between the measured and predicted values, and ModelESS is the fitting index of the model.

	MC [%]	D50 [μm]	TD [kg/m^3^]	HR
*SEP*	1.66	25.1	42.0	0.0721
*PRESS*	96.2	22018	61639	0.182
*γ* ^2^	0.985	−0.115	0.823	0.873
*LV*	5	4	4	3
*Slope*	0.928	−0.138	0.833	0.554
*Intercept*	−0.515	180	110	0.528
*ModelESS*	225.0	68.7	257.2	70.0
PTM	SNV	NOR	SNV	NOR

## Data Availability

Data is contained within the article.

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
