# Peer review of "Real-Time Monitoring of Critical Quality Attributes during High-Shear Wet Granulation Process by Near-Infrared Spectroscopy Effect of Water Addition and Stirring Speed on Pharmaceutical Properties of the Granules"

_pharmaceuticals, 2022, doi:10.3390/ph15070822_

Round 1
Reviewer 1 Report
The authors used partial least squares regression calibration models to predict the pharmaceutical properties of granules prepared using HSWG and studied the effects of the water amount, water addition rate, and impeller agitation speed on the pharmaceutical properties of the granules. The paper is well prepared, and its idea is innovative. I recommend the publication of the paper after minor revision: Tables are presented as print screen. All the figures have low resolution. What are the pharmaceutical applications of HSWG? A nomenclature is to be added. More details on the prediction technique are to be added. Are there any standards related to the production of HSWG granules?
Author Response
Thank you very much for your great evaluation for our paper.
The response opinion was enclosed as the comment file.
Reviewer 2 Report
Apart from minor editorial shortcomings, I do not see any major substantive flaws.
Author Response

(The authors gave the same response as above.)

Reviewer 3 Report
General comments
The topic of the paper is of interest and highly relevant in view of current efforts within digitalization in the pharmaceutical industry. The paper is well -written, with only a few minor adjustments to be made, See the specific comments below. The discussion of the results is quite extensive, which is nice for such an experimental paper.
Specific comments
- Line 14: You need to define the term auto-manufacture! It appears twice in the paper but you do not explain what you mean with that term. I have never heard that term before, by the way. Check the relevant literature and use the relevant terminology.
- Line 22-23: This sentence needs to be rephrased. It could for example be: “The temporal changes in the pharmaceutical properties observed for different amounts of water added and for a range of stirring speed values were …”
- Line 39: The term “skilled veteran technician" should be replaced by another term. For example “experienced technician”
- Line 59: “… by feedback of the process control information” is an expression that makes not too much sense. I propose to simply remove it from this paper.
- Line 62: Should “reflectivity” not be replaced by ‘reflectance’?
- Line 116: What do you mean when you write “corrected sample granules”?
- Line 235 – 236: Instead of writing that the result was slightly complicated, I have the feeling that you should provide some more explanation in the text, specifically at this point!
- Line 241: You need to provide a better explanation why the values of experiment P9 were excluded!
- Line 260-261: You wrote that “The TD of the HSWG granules decreased with increasing MC, and the D50/MC profiles”. But is MC not proportional with the amount of water added? I would then expect the TD to increase with increased water addition, similar to what is stated in lines 238-239 of this paper.
- Figure 2: What does the color scale mean? Does it somehow illustrate the evolution of the NIR signal as a function of time? That is not very clear
Author Response

(The authors gave the same response as above.)
